# Demonstration of a Modular Prototype End-to-End Simulator for Aquatic Remote Sensing Applications

**DOI:** 10.3390/s23187824

**Published:** 2023-09-12

**Authors:** Mark W. Matthews, Arnold Dekker, Ian Price, Nathan Drayson, Joshua Pease, David Antoine, Janet Anstee, Robert Sharp, William Woodgate, Stuart Phinn, Stephen Gensemer

**Affiliations:** 1CyanoLakes (Pty) Ltd., Sydney, NSW 2126, Australia; mark@cyanolakes.com; 2Satdek (Pty) Ltd., Sutton, NSW 2620, Australia; 3CSIRO Space and Astronomy, Canberra, ACT 2601, Australia; 4Research School of Astronomy and Astrophysics, College of Science, Australian National University, Canberra, ACT 2601, Australia; 5CSIRO Environment, Canberra, ACT 2601, Australia; 6CSIRO Manufacturing, Melbourne, VIC 3216, Australia; 7Remote Sensing and Satellite Research Group, School of Earth and Planetary Sciences, Curtin University, Perth, WA 6845, Australia; 8School of the Environment, The University of Queensland, St Lucia Campus, St Lucia, QLD 4067, Australia

**Keywords:** end-to-end simulator, optical sensors, design, remote sensing, optics, satellite, SmallSat, CubeSat, coral reefs, cyanobacterial blooms, bathymetry

## Abstract

This study introduces a prototype end-to-end Simulator software tool for simulating two-dimensional satellite multispectral imagery for a variety of satellite instrument models in aquatic environments. Using case studies, the impact of variable sensor configurations on the performance of value-added products for challenging applications, such as coral reefs and cyanobacterial algal blooms, is assessed. This demonstrates how decisions regarding satellite sensor design, driven by cost constraints, directly influence the quality of value-added remote sensing products. Furthermore, the Simulator is used to identify situations where retrieval algorithms require further parameterization before application to unsimulated satellite data, where error sources cannot always be identified or isolated. The application of the Simulator can verify whether a given instrument design meets the performance requirements of end-users before build and launch, critically allowing for the justification of the cost and specifications for planned and future sensors. It is hoped that the Simulator will enable engineers and scientists to understand important design trade-offs in phase 0/A studies easily, quickly, reliably, and accurately in future Earth observation satellites and systems.

## 1. Introduction

AquaWatch Australia is a mission of the Commonwealth Science and Industrial Research Organisation (CSIRO, Australia’s national science agency) to build an integrated, operational system combining in situ and Earth observation data for monitoring and managing Australia’s inland and coastal water bodies. The AquaWatch Concurrent Design Facility study [1] identified the need for an end-to-end simulator that could assess trade-offs in satellite instrument design in early phase 0/A studies. This led to the present study called the “AquaWatch Pathfinders: Earth Observation (EO) Sensor Design Simulator Testbed (End-to-End Simulator)”. A virtual testbed in the form of an end-to-end simulation tool is an essential component of the design process of any new Earth imaging system [2,3,4,5,6]. The development of new optical satellite sensors requires multiple trade-offs between cost and sensor specifications, including the ground sampling distance (GSD); the number, width, and position of spectral bands; swath width and the instantaneous field of view; temporal resolution (revisit time); and radiometric sensitivity. New sensors must demonstrate that they meet or exceed the minimum requirements for a given application, and the impact of any sensor design trade-off on the quality of the final value-added product should be clearly demonstrated to assess the benefits of costly design decisions. Having a tool that can simulate how sensor design trade-offs impact the final product quality will enhance the efficient and cost-effective design of future sensors. Sensor design trade-offs are usually assessed by using airborne or in situ remote sensing data [7], which are processed using an instrument model or from highly specialized end-to-end simulators [8,9,10,11,12]. Having a sensor-agnostic or sensor-flexible simulator tool minimizes the need to collect airborne data and allows for greater flexibility in exploring various applications. 

An end-to-end simulator must capture the entire process of satellite observation, including the forward (observation using the sensor) and inverse (producing value-added products from the sensor) components (Figure 1). Briefly, the forward component incorporates the simulation of water-leaving radiance, propagation through the atmosphere, and the sampling of a scene using a sensor model to produce sensor images of raw uncalibrated radiance. The sensor model sampling the scene (across the track and along the track) incorporates the spatial, spectral, and radiometric characteristics of the sensor. The inverse problem includes the calibration of at-sensor radiances (so-called *Level 1* processing), georeferencing, atmospheric correction to produce remote sensing reflectance (*R_rs_*) (see Appendix A for definition) values, and the application of algorithms to produce quantitative geophysical value-added products (*Level 2* processing). 

This study presents the initial results from a modular prototype end-to-end simulator (the Simulator) similar to the models in [5,8,12], which have been designed for and applied to water applications. As aquatic applications have more demanding instrument requirements than typical vegetation applications (e.g., higher signal-to-noise ratio, narrow spectral bands) [13,14], design decisions have greater consequences for product quality [2,5,12]. The Simulator can simulate two-dimensional images for various satellite instrument model configurations using simulated input radiance data from synthetically constructed scenes to investigate the impact on value-added products. Using the output from in-water simulations as input data to the Simulator, the effect of sensor configuration on value-added products (e.g., chlorophyll-*a* (chl-*a*); water depth (bathymetry); and substrate type, which refers to the type of material on the bottom of the lake or ocean floor such as sand, coral, or seagrass) can be directly assessed. Similarly, differences in water quality information products due to the application of a range of retrieval algorithms can also be determined. Several case studies are presented, along with a simplified trade-off study, to demonstrate the utility of the prototype Simulator as an instrument design tool for examining trade-offs in instrument specification and instrument performance for a variety of water-related applications. This study explores the resulting differences in product quality for CubeSat versus SmallSat class instruments (e.g., [15,16,17]), thereby enabling a quantitative assessment of the impact of design decisions on downstream applications.

## 2. Materials and Methods

### 2.1. Simulator Description

#### 2.1.1. Architecture and Scene Construction

The prototype Simulator supports a set of instrument types, each with a suite of instrument parameters that may be set by the user. This is the basis for exploring various satellite instrument designs. The instrument samples the radiance field at the top of the atmosphere as it transits a region of Earth’s surface and generates the corresponding *Level-0* data (uncalibrated digital numbers). The radiance across the region is constructed from 2D geospatial maps and 1D hemispherical radiance fields. This leverages 1D radiative transfer simulations to provide a pseudo-3D description of spatially varying TOA radiance over large regions. The instrument samples this region on demand, based on the position and orientation of the instrument as it orbits Earth. The pseudo-3D structure is defined as a “*scene*”. It is a flexible model that can describe spatially varying spectral radiance over large areas with a very small quantity of input data, but the inherent flexibility demands complex structures to define the input.

The goal of the scene is to allow the user to construct models that represent realistic scenarios that an actual satellite-borne instrument would observe. A library of 1D radiative transfer simulations provides a multidimensional parametric space for reflectance (Figure 2). 

Spatial maps describe the variation in reflectance at Earth’s surface. A scene description file merges all of this information into a structure that allows the spectral radiance in any direction at any location to be extracted. With this scene description language, the user can construct a model of the reflectance with a variation that is applicable to that environment (e.g., Figure 3). The Simulator translates this human-readable description into a software object representing the entire scene and allows the instrument to sample the scene on demand. The spatial units in scene construction and sampling are arbitrary and therefore may not represent feature sizes in nature. The relative motion of the satellite and the optical configuration determine precisely where the scene is sampled and the direction of the radiance incident on the sensor elements within the instrument.

#### 2.1.2. Instrument Models and Calibration 

The Simulator has instrument models for different types of instruments implemented in the code. These software models translate spectral radiance at the entrance pupil of the instrument through to the raw digital sample of the imaging array sensor. The instrument models are parametric, allowing a wealth of physical properties to be specified, and the Simulator extracts them from text values in a user-supplied input file. This allows the effects of instrument design changes to be explored. Properties common to many instruments include optical aperture size, sensor pixel size, sensor read noise, sensor pixel well depth, sensor gain, optical spectral transmission efficiency functions, sensor quantum efficiency function, analog to digital converter bits-per-pixel, and amplifier bias level. One of the instrument models is based on a long-slit spectrograph with an internal dispersive optical element, such as a prism or diffraction grating, analogous to the pathfinder “Compact Hyperspectral Imager for the Coastal Ocean” or “CHICO” (Australian National University). Another is based on direct imaging with a gradient index filter, similar to the pathfinder “CyanoSat” (CSIRO) [18]. In each case, the software models the overall effect of the optical train, not the individual optical elements. 

The Simulator incorporates two idealized instrument models: one long-slit spectrograph that functions as an ideal radiometer in every sensor pixel, and a second multispectral instrument with user-defined band-pass filters. This allows for comparisons with the spectral band configurations of other satellite sensors, such as the Sentinel-2 Multispectral Imager [19] and the Sentinel-3 Ocean and Land Color Instrument (OLCI) [20]. However, the configuration is not necessarily something that could be physically realized. Modeling the characteristics of imaging sensors is common to most of these specification instruments. The Simulator is designed for extension, by software development in an object-oriented paradigm, to include more instrument models. It is anticipated that as an instrument design matures, a more elaborate software model will be developed. One of the existing instrument models would likely provide the starting point. Although a ray-tracing approach would be possible, it is not what is intended. Decomposing the transformation into a series of steps and incorporating additional models of the physical processes at the intermediate steps is more appropriate.

The objective of the Simulator is to provide realistic raw data so that the consequences of design decisions can be explored. The calibration of raw data is a critical part of this assessment. The Simulator essentially generates *Level-0* data products (uncalibrated digital numbers). It can also be used to produce the auxiliary data needed to calibrate the instrument. A utility program was developed to preprocess calibration data and transform *Level-0* products to *Level-1* products (calibrated radiances). This transformation results in a measure of the TOA radiance (*L_TOA_*), incorporating the effects of instrument noise in the raw data and calibration data. It allows the consequences of both random and systematic errors in calibrating the instrument to be explored (see Appendix A for details on calibration).

### 2.2. Demonstration of the Simulator

#### 2.2.1. Case Studies

Input datasets were generated for four applications: the surveillance of cyanobacterial blooms in small freshwater reservoirs; the mapping of shallow coral reefs; monitoring turbidity in aquaculture operations in shallow coastal waters; and monitoring chl-*a* and CDOM in reservoirs for drinking water supply. Here, we present two of these case studies. The first case study is of a hypothetical cyanobacterial bloom representative of Lake Hume, a large drinking and agricultural water reservoir located in southeastern Australia [21]. The IOPs and *R_rs_* at Lake Hume have been characterized by previous field measurements [22]. Here, we demonstrate the retrieval of chl-*a* pigment. This case study was also used to assess the trade-off between the SmallSat and CubeSat class instruments. The second case study represents idealized shallow-water seagrass mapping at Heron Island based on a spectral library of substratum reflectance features from seagrass and corals that have been measured at the site [23]. For this case, the signal from the water column was removed as noise to determine the fractional composition of the substrate (ocean floor), as well as chl-*a* and water depth.

#### 2.2.2. Input Datasets and Scene Generation

Forward radiative transfer models capable of modeling the light field are required to generate the input data required by the Simulator to construct a hypothetical scene that is sampled using a given instrument model. The radiative transfer code selected for this study was the open source Ocean Successive Orders with Atmosphere—Advanced (OSOAA) [24]. Coupled water–atmosphere radiative transfer simulations with OSOAA were used to determine *L_TOA_*, path radiances (*L_path_*), upward atmospheric transmittance (*T_up_*), and surface irradiances (*E_d_*0) for a range of water and atmosphere conditions for the relevant wavelengths and geometries. 

The Heron Island scene was modeled as optically shallow with a single set of SIOPs, while the water depth varied between one and seven meters with three bottom types: coral, sand, and seagrass. Bottom reflectance data and SIOPs collected from Heron Island in 2018 were used [23]. The hypothetical Lake Hume scene was constructed from *L_TOA_* modeled using a single set of SIOPs with variable chl-*a* concentrations (1 to 100 µg/L). In both cases, an aerosol model was selected for the atmosphere (maritime model for Heron Island and continental for Lake Hume) [25]. An aerosol optical thickness of 0.1 at 550 nm was chosen. A Gaussian random field generator [26] was used to represent the distribution of chl-*a* concentrations in a 3000 × 2000 array. For further details, see Appendix B. 

#### 2.2.3. Hypothetical Satellite Instruments

A generic multispectral sensor was specified with a nominal aperture size of 120 mm and spectral bands based on OLCI. OLCI bands were used because they are ideally positioned for water applications and readily allow for the application of existing algorithms [20]. For this simplified design trade-off study, we considered two additional hypothetical instrument configurations: A “CubeSat” class instrument with a 60 mm, f#-2.75 telescope utilizing a small-pixel focal-plane array;A “SmallSat” class instrument with a 240 mm f#-1.80 telescope utilizing a large-pixel focal-plane array with an increased full-well capacity and gain.

A full list of instrument design parameters is given in Table 1. To avoid adding unnecessary complexity to the results, the spectral band configuration and response functions of OLCI were used. To avoid complexity, the sampling resolution and spectral configuration were held constant, and only differences in instrument noise resulting from different aperture configurations were explored. 

#### 2.2.4. Inversion and Validation 

TOA radiances obtained as output from the calibration of the instrument model were atmospherically corrected to determine *R_rs_* using ancillary data products generated from the radiative transfer simulations used as input to the instrument model (see Appendix A for details). The aim of this exact atmospheric correction was not to test the performance of any specific algorithm but rather to assess the differences in product quality introduced solely from variable instrument design configurations. For the Heron Island case, the SAMBUCA (semi-analytical model for bathymetry, unmixing, and concentration assessment) algorithm [27] was used to estimate water depth, as well as the bottom type from the atmospherically corrected scenes. SAMBUCA simultaneously retrieves the concentration of chl-*a* and non-algal particles (NAPs), the absorption by colored dissolved organic matter ($a_{CDOM}$), the percent substratum (bottom) cover, and uncertainty estimates via a substrate detectability index. SAMBUCA was parameterized using the same dataset used to produce the input dataset, including SIOPs and benthic spectra for *Acropora* sp. coal, coralline sand, and turf algae commonly found at Heron Island. The advanced linear matrix inversion (aLMI) [28] was used to estimate chl-*a*, NAPs, and $a_{cdom}$ from the *R_rs_* scenes for the Lake Hume case. Then, aLMI was parameterized with the same data used to produce the input datasets and provided with a set of eight averaged SIOP sets that capture the range of variability in NAPs and CDOM based on in situ measurements at Lake Hume. Again, the aim was not to assess the performance of different algorithms but rather to examine how variable instrument design and algorithm retrieval affected product quality. Performance of value-added product

Retrievals using SAMBUCA and aLMI were evaluated with mean absolute error (*MAE*) and bias calculated in log space after [29]:(1)MAE=10^∑i=1n|log10(Mi−log10(Oi)|n
(2)bias=10^∑i=1nlog10(Mi−log10Oin
where *M*, *O*, and *n* are the modeled, observed value, and the sample size, respectively. 

## 3. Results

### 3.1. Water Depth and Chl-a Retrieval Applications

The inputs and outputs for water depth and chl-*a* value-added products for the hypothetical Heron Island and Lake Hume and scenes for a generic multispectral satellite instrument are shown in Figure 4. The output scenes show the introduction of noise from the instrument model, as well as uncertainties introduced through the applications of retrieval algorithms. This demonstrates the process of observing a target using a satellite sensor, which inherently introduces random noise that affects the overall quality of parameter retrievals. This is especially evident in the depth map retrieved using SAMBUCA, which shows patches of deep water retrieved using the model that are not present in the input data. These erroneous depths were highly correlated with incorrect *Acropora* sp. coral substrate identifications, which correlated to the location of seagrass in the input scenes. It is evident that the substrate type selected for algorithm parameterization contributed to erroneous depth estimations. This demonstrates how the Simulator can be used to identify cases where algorithms require further training or more representative parameterization before they are applied to unsimulated satellite data, where error sources cannot always be identified or constrained.

A comparison of pixel-for-pixel chl-*a* and depth retrievals (Figure 5) validates the parameter retrievals from the algorithms against the original simulated inputs. This quantitatively demonstrates how the noise introduced through instrument sampling and inversion process affects parameter retrieval performance. For complex water cases (such as the Heron Island case explored here) and complex products (such as substrate type), there will always be some discrepancy between inputs and outputs. In this case, the Simulator provides engineers and scientists with quantitative (best) performance estimates that arise solely from instrument noise and retrieval algorithms, enabling the quantification of these effects before satellites are built or launched. Based on the satellite algorithm configurations presented here, the best-case MAE for chl-*a* and water depth estimates over complex cyanobacterial bloom and reef systems was 14% and 87%, respectively. While over- and underestimations were evenly balanced for chl-*a*, depth was strongly overestimated in regions dominated by seagrass. This highlights how the Simulator can be used to assess trade-offs in instrument design that lead to better or worse product outcomes, as demonstrated in the next section.

### 3.2. CubeSat versus SmallSat Trade-Off Demonstration

The effect of instrument aperture size (60 vs. 240 mm, CubeSat versus SmallSat) on the retrieval quality of chl-*a* concentration for the synthetic Lake Hume cyanobacterial bloom case is shown in Figure 6 and Figure 7. The results show the quantitative differences in product performance that can be expected between a CubeSat and SmallSat satellite instrument. The smaller aperture size results in increased image speckling and over- and underestimation for chl-*a* (visible in Figure 6 but more clearly seen in Figure 7). The imager with the larger aperture resulted in a value-added product with significantly higher definition, less speckling, and smaller errors than the smaller CubeSat configuration. Figure 7 indicates the significant influence that instrument design had on inversion performance and product accuracy—the only difference in this case being from the instrument aperture size as the algorithms were identical. aLMI retrievals from the CubeSat imager required a greater number of SIOP sets to find a solution (all eight sets were used) than the retrievals using the simulated SmallSat image (only three SIOP sets were required for the SmallSat image). Given that a single SIOP set was used in scene creation, this indicates that the poorer optical performance of the CubeSat imager led to inappropriate SIOP selection, which contributed to poor inversion performance. Furthermore, 12,748 fewer pixels were returned by aLMI using the CubeSat imager. This lower return rate indicates that some pixels could not be accurately evaluated using the model and were therefore discarded. The aLMI retrievals from the larger aperture SmallSat imager had lower MAE for chl-*a* (11%) and narrower retrieval distributions for NAP concentration. In both cases, the retrieval of CDOM absorption was poor; however, this was expected in the case of the simulated high-biomass cyanobacterial bloom. 

## 4. Discussion

This study demonstrates a prototype end-to-end simulator for informing satellite instrument design for aquatic applications. This includes water and atmosphere forward models to generate input data, software that enables hypothetical scene creation and sampling using an instrument model, and the inversion process used to create and validate value-added products from raw data from the top of the atmosphere. This simplified design trade-off study demonstrates how design decisions, directly driven by cost-related constraints, directly influence product resolution and quality. Thus, the Simulator is a suitable tool for investigating return on investment and user requirements in terms of product performance without requiring the collection of airborne datasets [5]. The consequences of design decisions can be thoroughly assessed before instruments are built and launched. In addition, the Simulator identified specific cases where algorithm retrieval was inaccurate, enabling algorithms to be fine-tuned and parameterized to improve performance before application on unsimulated satellite data, where error sources cannot always be identified or constrained [10]. The simple instrument trade-off example demonstrates the differences that can be expected in end-user product quality resulting from instrument design decisions that are usually driven by cost. It shows how the Simulator can be used to directly verify whether a given instrument design or configuration meets the accuracy or performance requirements of end-users before build and launch. The modular nature of the Simulator enables a quick examination of spectral, spatial, and radiometric trade-offs without requiring highly complex simulators that are only applicable to a particular mission (see below). The comparison between the SmallSat and CubeSat instruments provides quantitative performance metrics to support the use of larger, more expensive satellite instruments for the end-user-driven application under consideration. This is a critical outcome in justifying mission specifications and costs for future and planned satellite missions. 

Similar software tools, already described above, are inadequate in that they are limited to a specific sensor, to the extent that they can no longer be used to effectively assess design trade-offs from multiple configurations early in the design process [5,7,8,9,10]. Such simulators include those for missions such as EnMAP [8,11], the Fluorescence Explorer (FLEX) [9], the Far-Infrared Outgoing Radiation Understanding and Monitoring (FORUM) [10] and Sentinel-3 [12]. The Selex Galileo is a commercial simulator similar to that described here, which is designed for early phase 0/A missions to enable the rapid dimensioning of new optical instruments and to trace the link to user requirements [5]. There are also several other commercially available simulators with various degrees of sophistication (see references in [5]). As the Simulator described here is a prototype, a detailed comparison with existing alternatives is likely premature. Rather, this paper demonstrates tangible outcomes related to sensor design affecting aquatic applications.

The prototype Simulator should be improved and developed further as a tool for both instrument design trade-off studies and for research and development applications. For aquatic applications, functionality implementing realistic adjacency effects (stray light from bright targets surrounding water) and sun glint on water surfaces should be prioritized, as in [12], which have been neglected in the cases shown here. The Simulator does not account for the impact of cloudy conditions (e.g., [10]). The implementation of advanced 3D radiative transfer models that simulate landscapes (e.g., [30,31]) not currently implemented in the prototype Simulator, has the potential to provide even more realistic simulations in the future [4]. Several improvements in the graphical user interface would facilitate easier use by engineers and scientists, similar to [32]. Further additional development of the prototype Simulator for applications related to terrestrial environments is envisaged that would include terrain geometries and bio-optical models for vegetation and land surface types as forward modeling components [6]. With further development, the prototype Simulator could be made available either commercially or open source [30,32] as a generalized, stable, complete, and user-friendly tool to support a broader range of EO applications, including ground sensing. This would help users understand important design trade-offs easily, quickly, reliably, and accurately in EO satellites and systems.

## Figures and Tables

**Figure 1 sensors-23-07824-f001:**
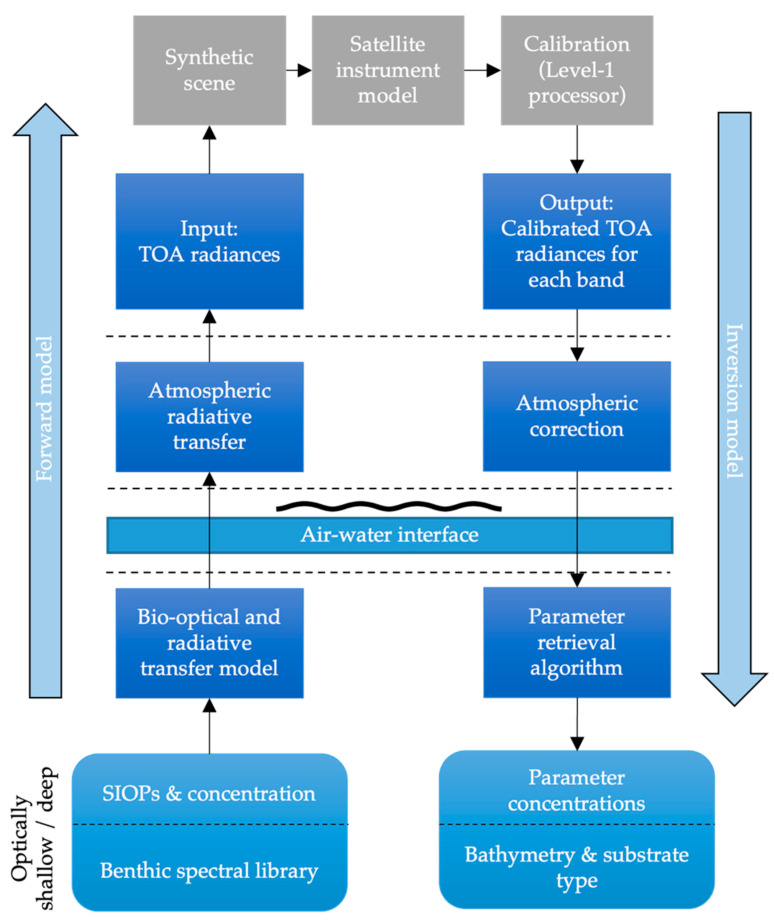
Flowchart showing the modular components of the Simulator, and forward and inverse processes. TOA = top of the atmosphere. Inherent optical properties (IOPs) refer to the absorption (*a*), scattering (*b*), and backscattering (*b_b_*) volume coefficients. The specific IOPs (SIOPs) refer to IOPs normalized by the concentration of a parameter, such as chl-*a*.

**Figure 2 sensors-23-07824-f002:**
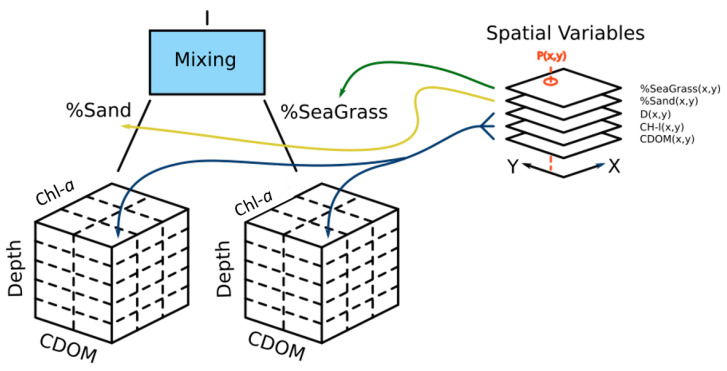
Schematic representation of the structure of a scene for a shallow aquatic environment. CDOM = colored dissolved organic matter.

**Figure 3 sensors-23-07824-f003:**
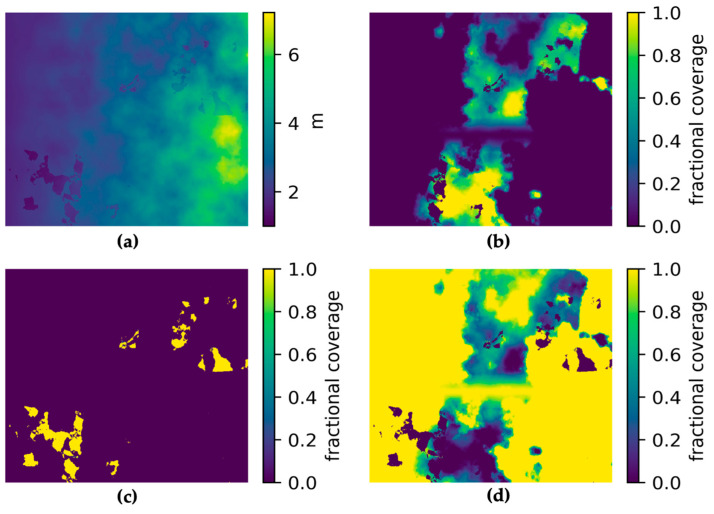
Scene construction for a synthetic coral reef: (**a**) water depth in meters; (**b**) fractional cover of seagrass in percent; (**c**) fractional coverage of rock in percent; (**d**) fractional coverage of sand in percent. Scene units are arbitrary.

**Figure 4 sensors-23-07824-f004:**
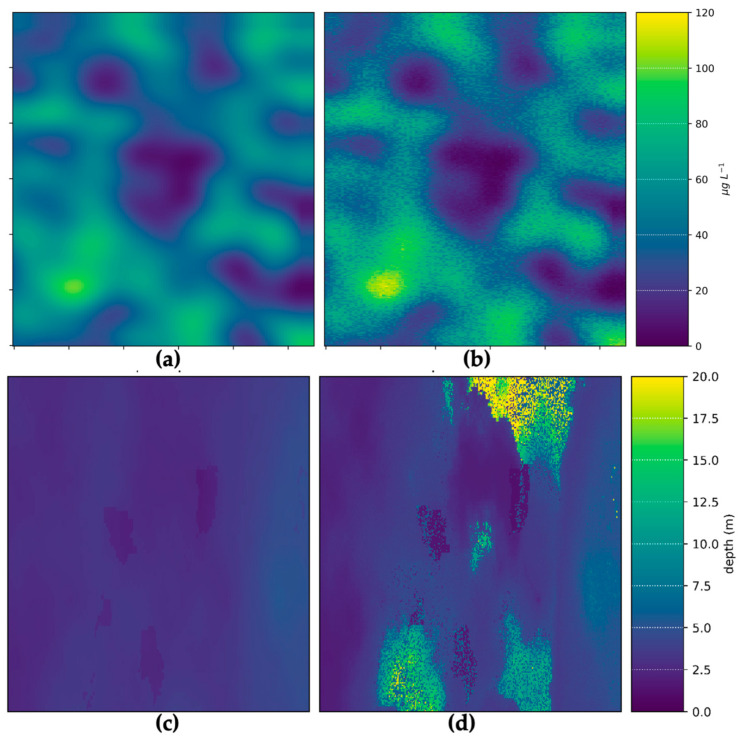
Comparison between the input scene and the simulated output for a generic multispectral sensor: (**a**) input chl-*a* map; (**b**) output chl-*a* map estimated using the aLMI algorithm; (**c**) input depth map; (**d**) estimated depth using the SAMBUCA algorithm. Scene units are arbitrary.

**Figure 5 sensors-23-07824-f005:**
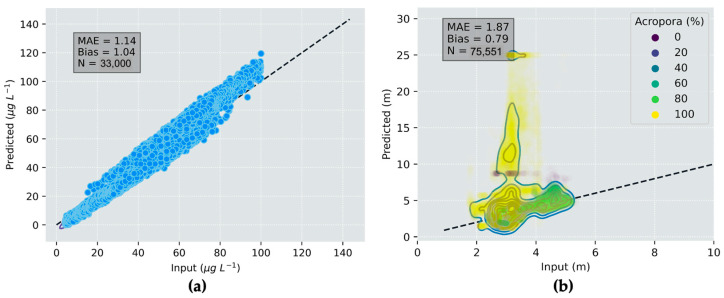
Comparison between the actual parameter values and those estimated using the output from a simulated sensor: (**a**) Chl-*a* for the Lake Hume scene estimated using the aLMI algorithm; (**b**) depth for the Heron Island scene estimated using the SAMBUCA algorithm. Colors in (**b**) indicate the fraction of the coral *Acropora* sp. in which substrate type influenced the estimation of depth.

**Figure 6 sensors-23-07824-f006:**
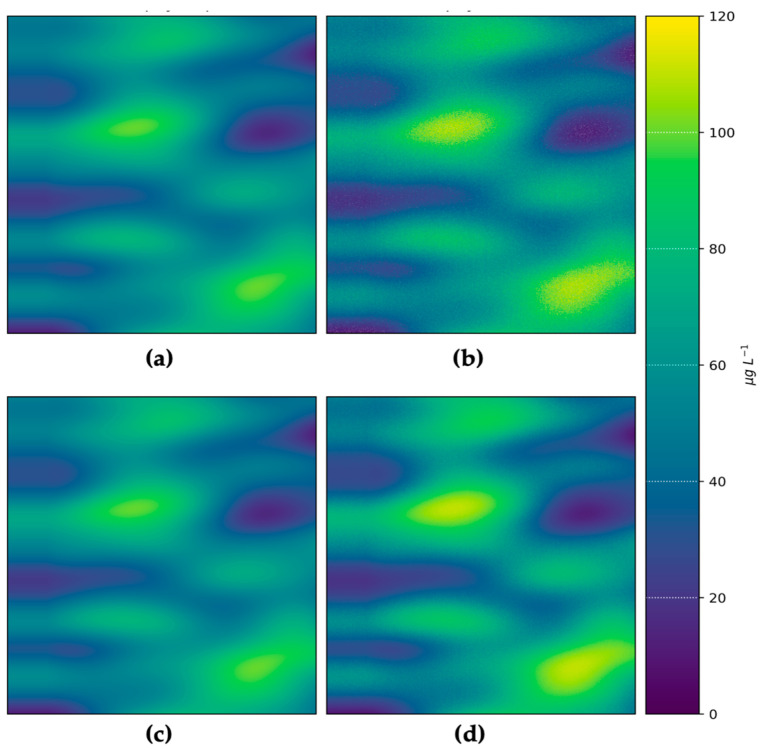
Chl-*a* maps for CubeSat and SmallSat instrument configurations estimated using the aLMI algorithm for a Lake Hume scene: (**a**) actual chl-*a*; (**b**) chl-*a* estimated based on a simulated CubeSat; (**c**) actual chl-*a*; (**d**) chl-*a* estimated based on a simulated SmallSat. Scene units are arbitrary.

**Figure 7 sensors-23-07824-f007:**
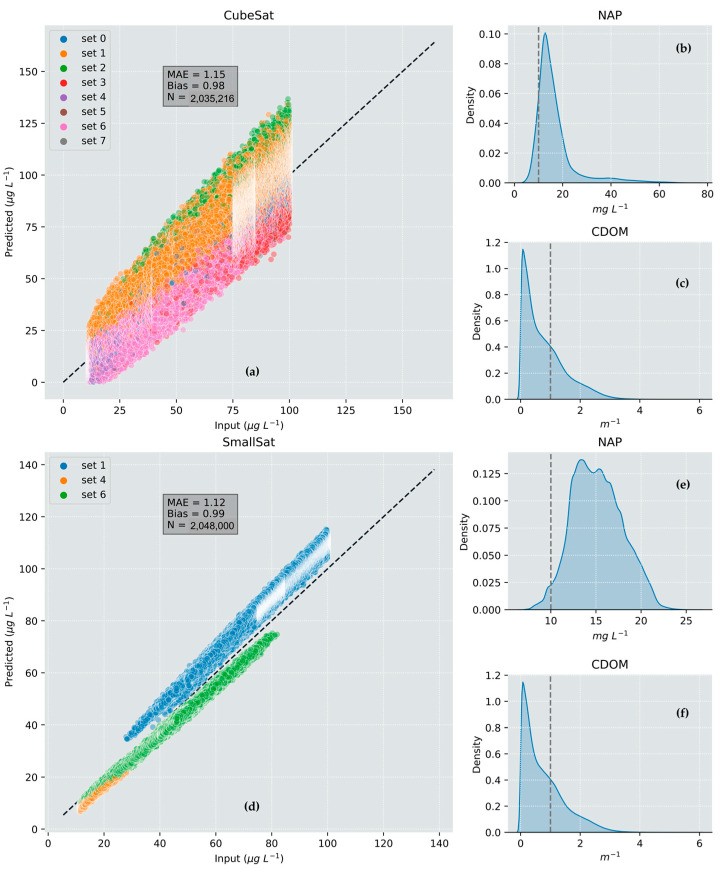
Comparison of product retrieval performance for simulated CubeSat and SmallSat instrument configurations. The colored legend shows different SIOP sets selected when solving using the aLMI algorithm. The dotted line on the histograms shows the actual constant values: (**a**) CubeSat chl-*a*; (**b**) CubeSat NAP; (**c**) CubeSat CDOM; (**d**) SmallSat chl-*a*; (**e**) SmallSat NAP; (**f**) SmallSat CDOM.

**Table 1 sensors-23-07824-t001:** Configurations of hypothetical sensor instruments.

	Feature	Nominal	CubeSat	SmallSat
Instrument geometry	Altitude (km)	560	560	560
	Orbital heading azimuth	0.0	0.0	0.0
Instrument Design	Polarimeter	None	None	None
	Aperture size	120 mm	60 mm	240 mm
	Pixel size	5.5 µm	5.86 µm	15.5 µm
	Focal length	165 mm	165 mm	432 mm
	Number of sensors	1024	1024	1024
	Exposure time	0.0015 s	0.0015 s	0.0015 s
	Number of exposures	1536	2000	2000
	Forward tilt angle	0	0	0
	Read noise	15.0 e	15.0 e	15.0 e
	Pixel well depth	80,000 e	80,000 e	640,000 e
	Gain	3.0 e/ADU	3.0 e/ADU	10 e/ADU
	Bias	10.4	10.4	10.4
	Bits	16	16	16
	Optical transmission	0.885	0.885	0.885
	Number of bands	19	19	19
	Centre wavelengths (nm)	400, 411.8, 442.9, 490.5, 510.5, 560.5, 620.4, 665.3, 674, 681.6, 709.1, 754.2, 761.7, 764.8, 767.9, 779.2, 865.4, 884.3, 897.4		
	Sampling resolution	20		
	Spectral response function	Sentinel 3A OLCI		

## Data Availability

Requests for the Simulator code and associated input data described in this report should be made to the SmartSat CRC (https://www.smartsatcrc.com/ (accessed on 11 September 2023). OSOAA is available from https://github.com/CNES/RadiativeTransferCode-OSOAA (accessed on 15 June 2022). The SAMBUCA algorithm is available from https://github.com/stevesagar/sambuca (accessed on 11 September 2023). The Australian Bio-Optical database is available from the CSIRO at https://doi.org/10.25919/rtd7-j815 (accessed on 11 September 2023). The aLMI algorithm is available from https://github.com/GeoscienceAustralia/DEA-Water-Quality (accessed on 11 September 2023).

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
