# Peer review of "Demonstration of a Modular Prototype End-to-End Simulator for Aquatic Remote Sensing Applications"

_sensors, 2023, doi:10.3390/s23187824_

Round 1
Reviewer 1 Report
Dear authors,
I attached the revised version with comments on the text.
In general, there are two main weaks points.
First point, there are very few references, many paragraphs without a single reference. I have annotated the manuscript at several point where I think references are necessary. However, I recommend the authors critically reading their manuscript and add references where needed.
Second point, the authors do not explain many terms such as IOPs, SIOPs, AOPs, CDOM, NAP, chla... These terms appear for instance in figures without even been mentioned in the text before. "Sensors" is not an oceanography or limnology journal, so it is important to define correctly these terms from the begining, and do a correct use of these terms. For instance, see line 217 "water constituent concentrations (chl-a)" this is not correct at all. Optical active compounds include Chl-a, CDOM and NAP, so this is not correct. We cannot say that chl-a included "all water constituents" because it is only representative of phytoplankon. So, it is important that the authors make a proper use of the specific vocabulary

Author Response
First point, there are very few references, many paragraphs without a single reference. I have annotated the manuscript at several point where I think references are necessary. However, I recommend the authors critically reading their manuscript and add references where needed.
Several references have been added where requested.
Second point, the authors do not explain many terms such as IOPs, SIOPs, AOPs, CDOM, NAP, chla... These terms appear for instance in figures without even been mentioned in the text before. "Sensors" is not an oceanography or limnology journal, so it is important to define correctly these terms from the begining, and do a correct use of these terms. For instance, see line 217 "water constituent concentrations (chl-a)" this is not correct at all. Optical active compounds include Chl-a, CDOM and NAP, so this is not correct. We cannot say that chl-a included "all water constituents" because it is only representative of phytoplankon. So, it is important that the authors make a proper use of the specific vocabulary.
Where possible, we have removed technical terms, e.g., AOPs. We have ensured all terms are defined in the manuscript in response to your comment.
L19: I wouldn't say that a cyanobacterial algal blooms is an aquatic system, the aquatic system could be a lake, an estuary, etc. and the bloom is transient conditions
Changed to “challenging applications”
L20: Too concise sentence, I don't understand, design of what? which product?
Revised to: “This demonstrates how decisions regarding satellite sensor design, driven by cost-constraints, directly influence the quality of value-added remote sensing products”
L42: necessitates
Changed to “requires”
L58: Specify directly water leaving radiances, this study is on aquatic ecosystems
Changed to “water-leaving radiance”
L64: ground
Changed to “remote sensing reflectance (Rrs) (see Appendix B for definition)”
L68: citation needed
Reference 2 has been cited, which details requirements for water-colour satellite sensors.
L88: There are elements in Figure 1 that have not been described or mentioned before the figure. It is necessary to explain them (SIOPs, substrate type, bathymetry, ..)
Changed to: “Inherent Optical Properties (IOPs) refers to the absorption (a), scattering (b) and backscattering (bb) volume coefficients. The specific-IOPs (SIOPs) refers to IOPs normalized by the concentration of a parameter, such as chl-a.”
Substrate type and bathymetry have been more clearly defined earlier in the text:
“Using the output from in-water simulations as input data to the Simulator, the effect of sensor configuration on value added products (e.g., chlorophyll-a (chl-a), water depth (bathymetry), and substrate type which refers to the type of material on the bottom of the lake or ocean floor, such as sand, coral or seagrass) can be directly assessed.”
L107: AOPs should be defined, this is not an oceanography or water journal and the audience may not know these terms. Also, citation needed. In general there are very few references.
Reference to AOPs has been removed in the text in favour of “reflectance”:
“A library of 1D radiative transfer simulations provides a multi-dimensional parametric space for reflectance (Figure 2).”
“Spatial maps describe the variation in reflectance at the Earth’s surface.”
“With this scene description language, the user can construct a model of the reflectance with variation that is applicable to that environment (e.g., Figure 3).”
L108: what does "CH-I" mean?
Added to figure caption: “CH-l refers to chl-a.”
L127: paragraph needs references
We have added three additional references in this paragraph. The first for the cyanosat imager, the second for Sentinel-2 MSI and the third for Sentinel-3 OLCI. As the initial sentence of the paragraph describes general engineering characteristics of satellite sensors, no reference has been added.
Matthews, M.W; Kravitz, J.A.; Pease, J.; Gensemer, S. Determining the spectral requirements for cyanobacteria detection for the CyanoSat hyperspectral imager with machine learning. Remote Sens.Submitted.
Drusch, M.; Del Bello, U.; Carlier, S.; Colin, O.; Fernandez, V.; Gascon, F.; Hoersch, B.; Isola, C.; Laberinti, P.; Martimort, P.; Meygret, A. Sentinel-2: ESA's optical high-resolution mission for GMES operational services.Remote Sens. Environ., 2012, 120, 25-36.
Nieke J.; Frerick J.; Stroede J.; Mavrocordatos, C.; Berruti, B. Status of the optical payload and processor development of ESA's Sentinel 3 mission. Int. Geosci. Remote Sens. Symp., 2008, IV, 427-430.
L137: We need references
References have been added for MSI and OLCI later in the paragraph.
Changed to: “The Simulator incorporates two idealized instrument models; one long-slit spectrograph that functions as an ideal radiometer in every sensor pixel, and a second multispectral instrument with user defined band-pass filters.”
L163: This sentence is not clear at all. What does shallow coastal water for aquaculture? is it monitoring these waters quality? the same with "drinking water supply reservoirs" is it monitoring water quality? monitoring cyanobacteria blooms? Please, note it is CIANOBACTERIA BLOOMS not cyanobacterial.
Revised to: “Input datasets were generated for four applications: surveillance of cyanobacterial blooms in small freshwater reservoirs; mapping of shallow coral reefs; monitoring turbidity in aquaculture operations in shallow coastal waters; and monitoring chl-a and CDOM is reservoirs for drinking water supply.” Some reviewers prefer cyanobacteria blooms, others have insisted it is cyanobacterial blooms, it seems to be a matter of preference.
L168: First time mentioned should be defined "chlorophyll"
This is defined at line 75.
L180: As I have said before, these properties should have been cited in the introduction to understand figure 1.
These have been clearly defined in the caption for Figure 1. Changed to “conditions”.
L185: which is the difference between IOPs and SIOPs, this should have been clarified from the begining of the manuscript
These have been clearly defined in the caption for Figure 1. Amended to “a single set of SIOPs”
L187: is this the micro??
Updated so that the symbol µ has been used throughout the manuscript.
L194: we need references
Reference has been added.
L217: please !! chlorophyll is not water constituent concentrations!! this is not correct
would the authors mean optically active constituents they should specify also NAP and CDOM
Sentence has been revised as follows:
“SAMBUCA simultaneously retrieves the concentration of chl-a and non-algal particles (NAP), the absorption by colored dissolved organic matter (a_cdom), the percent substratum (bottom) cover, and uncertainty estimates via a substrate detectability index.”
These terms have now been abbreviated where referenced on line 276.
Reviewer 2 Report
An end-to-end simulator could capture the entire process of satellite observation, which is beneficial for assessing the impact of sensor design on the final product quality. This study presents initial results from a modular end-to-end simulator for processing two-dimensional satellite multi-spectral imagery for a variety of satellite instrument models over aquatic environment. This topic is interesting and the tool is useful. However, some contents and figures are not so easy to read.
In order to improve the quality of this manuscript to meet the requirement of this journal, I suggest authors to pay attention to the following points.
1. Some figure captions are not so clear. Specific parameters could be shown in Fig. 3-7.
2. Figure 6. Differences between Fig.6a-b and Fig.6c-d are not obvious. How can we understand the results listed between Line 275-283? And What’s the reason?
3. Figure 7. What are the differences for sets 0-7? More information could be added.
4. Brief introductions of these case studies will be helpful for readers to understand the comparison results. Like Fig.4, how about the research background of Heron Island and Lake Hume?
5. How about the performance of this simulator compared with existing ones (like the EnMAP)? What’s the difference? And how about the advantage of the simulator tool in this study?
Author Response
An end-to-end simulator could capture the entire process of satellite observation, which is beneficial for assessing the impact of sensor design on the final product quality. This study presents initial results from a modular end-to-end simulator for processing two-dimensional satellite multi-spectral imagery for a variety of satellite instrument models over aquatic environment. This topic is interesting and the tool is useful. However, some contents and figures are not so easy to read. In order to improve the quality of this manuscript to meet the requirement of this journal, I suggest authors to pay attention to the following points.
- Some figure captions are not so clear. Specific parameters could be shown in Fig. 3-7.
Figure captions have been updated, corrected and clarified as follows:
Figure 3. Scene construction for a synthetic coral reef. (a) Water depth in meters; (b) fractional cover of seagrass in percent; (c) fractional coverage of rock in percent; (d) fractional coverage of sand in percent.
Figure 4. Comparison between the input scene and the simulated output for a generic multi-spectral sensor. (a) input chl-a map; (b) output chl-a map estimated using the aLMI algorithm; (c) input depth map; (d)estimated depth using the SAMBUCA algorithm.
Figure 5. Comparison between the actual parameter values and those estimated using the output from a simulated sensor. (a) Chl-a for Lake Hume scene estimated using the aLMI algorithm; (b) depth for Heron Island scene estimated using the SAMBUCA algorithm. Colors in (b) indicate fraction of the coral Acropora sp.that substrate type influenced the estimation of depth.
Figure 6. Chl-a maps for cubesat and smallsat instrument configurations estimated using the aLMI algorithm for a Lake Hume scene. (a) actual chl-a; (b) chl-a estimated based on a simulated cubesat; (c) actual chl-a; (d) chl-a estimated based on a simulated smallsat.
Figure 7. Comparison of product retrieval performance for simulated cubesat and smallsat instrument configurations. The colored legend show different SIOP sets selected when solving using the aLMI algorithm. The dotted line on the histograms shows the actual constant values. (a) Cubesat chl-a; (b) cubesat NAP; (c) cubesat CDOM; (d) smallsat chl-a; (e) smallsat NAP; (f) smallsat CDOM.
- Figure 6. Differences between Fig.6a-b and Fig.6c-d are not obvious. How can we understand the results listed between Line 275-283? And What’s the reason?
The reason the differences in Fig 6 are not obvious is probably because of the figure resolution, differences in screens, and perhaps eyesight even (!). It is hard to see the differences, but the quantitative results from Figure 6 shown in Figure 7 are easy to see. I have added the following sentence to clarify what the reader should look for:
"The results show the quantitative differences in product performance that can be expected between a cubesat and smallsat satellite instrument. The smaller aperture size results in increased image speckling and over and under-estimation for chl-a (visible in Fig. 6 but more clearly seen in Fig. 7)."
- Figure 7. What are the differences for sets 0-7? More information could be added.
We have added a description of the seven SIOP sets to section 2.2.3 methods:
“aLMI was parameterized using the same data used to produce the input datasets and provided with a set of eight averaged SIOPs sets that capture the range of variability in NAP and CDOM based on in situ measurements at Lake Hume.”
- Brief introductions of these case studies will be helpful for readers to understand the comparison results. Like Fig.4, how about the research background of Heron Island and Lake Hume?
Section 2.2.1 contains some references to previous work at Lake Hume and Heron Island. The following has been added to provide some additional background on research done there:
“The first case study is of a hypothetical cyanobacterial bloom representative of Lake Hume, a large drinking and agricultural water reservoir located in south-eastern Australia. The IOPs and Rrs at Lake Hume have been characterized by previous field measurements [7].”
“The second case study represents idealized shallow-water seagrass mapping at Heron Island based on spectral library of substratum reflectance features from seagrass and corals that have been measured at the site [8]”
- How about the performance of this simulator compared with existing ones (like the EnMAP)? What’s the difference? And how about the advantage of the simulator tool in this study?
As described in the text, Simulators become very specific for a mission, and therefore are generally not publicly available, so direct comparison is not possible. The main difference and advantage are that this a generic tool that can be customized and used across different scenarios for initial trade studies. This is explained in the discussion:
“The modular nature of the Simulator enables quick examination of spectral, spatial, and radiometric trade-offs without requiring highly complex simulators that are only applicable for a particular mission [e.g., 5].”
Reviewer 3 Report
The manuscript presents a study on a prototype end-to-end simulator for satellite multi-spectral imagery and several instrument models for remote sensing of aquatic environments. The paper is clear although some revision is needed to improve some sections (see my comments below).
General comment:
- Is there the possibility to test the simulator with independent data set?
- I do not understand why ‘Simulator’ is written with the capitol letter.
- The use of OLCI should be better introduced (Sentinel-2 MSI is also mentioned, while not Landsat. In any case as MSI is juts mentioned here without justification you could also remove it).
Line-by-line comments
Abstract: ‘It is hoped that the Simulator will enable engineers and scientists to understand important design trade-offs easily, quickly, reliably, and accurately in future Earth observation satellites and systems.’
Could you clarify on how this will be ensured? Are you going to share the code?
Line 51: ‘are usually assessed by collecting airborne remote sensing data [3]’: I do not fully agree as many E2E work also with in situ data and/or modeled data in addition to airborne imagery; please revise
Figure 1 and related text: it is not fully clear what authors intend for ‘Bio-optical and radiative transfer model’: what about ‘water radiative transfer model’? By keeping the two terms (i.e. bio-optical and radiative transfer model) it is unclear to what this box is referring to? (e.g. is a RT similar to Hydrolight? Is it a combination with SAMBUCA?); please clarify.
Figure 2: the quantity CH-I is not defined. Please explains all symbols by revising the text related to this figure.
Figure 4: could you add a scale bare to show the mapped areas by the simulator?
Lake Hume and Heron Island: how the 100 m pixel size might affect the simulator performance? The spatial resolution seems too coarse for both applications; or: which is the pixel size of maps in Fig.4?
Author Response
The manuscript presents a study on a prototype end-to-end simulator for satellite multi-spectral imagery and several instrument models for remote sensing of aquatic environments. The paper is clear although some revision is needed to improve some sections (see my comments below).
General comment:
- Is there the possibility to test the simulator with independent data set?
I guess you may be referring to validation of products or TOA radiances? The radiative transfer codes outputs have been validated elsewhere (see OSOAA) and given that these are synesthetic scenes they cannot be independently verified or compared to airborne or in situ radiance datasets. Thus, it is not really feasible to verify the output with an independent dataset. However, it could be compared to outputs from other simulators (if they were publicly available, which they are not). This is a preliminary study and certainly additional independent verification would be required later.
- I do not understand why ‘Simulator’ is written with the capitol letter.
This indicates where we refer to the specific Simulator (capital S) in this study, not a general simulator (small s) from elsewhere.
- The use of OLCI should be better introduced (Sentinel-2 MSI is also mentioned, while not Landsat. In any case as MSI is juts mentioned here without justification you could also remove it).
We have added a reference for OLCI: “OLCI bands were used because they are ideally positioned for water applications and readily allowed application of existing algorithms [20].” It makes sense to keep the reference to S2 because it is one of the configurations available in the simulator.
Line-by-line comments
Abstract: ‘It is hoped that the Simulator will enable engineers and scientists to understand important design trade-offs easily, quickly, reliably, and accurately in future Earth observation satellites and systems.’
Could you clarify on how this will be ensured? Are you going to share the code?
Sadly, there is no way to ensure anyone will use this tool. If it were up to me (first author) the code would already be public. However, the code is available on request from SmartSat who will evaluate each request on its merit (see data availability statement):
“Requests for the Simulator code and associated input data described in this report should be made to the SmartSat CRC”.
Line 51: ‘are usually assessed by collecting airborne remote sensing data [3]’: I do not fully agree as many E2E work also with in situ data and/or modeled data in addition to airborne imagery; please revise
Revised as follows: “Sensor design trade-offs are usually assessed by using airborne or in situ remote sensing data”
Figure 1 and related text: it is not fully clear what authors intend for ‘Bio-optical and radiative transfer model’: what about ‘water radiative transfer model’? By keeping the two terms (i.e. bio-optical and radiative transfer model) it is unclear to what this box is referring to? (e.g. is a RT similar to Hydrolight? Is it a combination with SAMBUCA?); please clarify.
The reason bio-optical and radiative transfer model are used separately, is because they are separate models. One is an additive bio-optical model producing the volume coefficients (a, b, bb, c, VSF), and the other is the radiative transfer code, calculating radiance and other parameters, such as Hydrolight. However, this could be simplified to “Water Radiative Transfer” if the reviewer insists.
Figure 2: the quantity CH-I is not defined. Please explains all symbols by revising the text related to this figure.
Added to figure caption: “CH-l refers to chl-a.”
Figure 4: could you add a scale bare to show the mapped areas by the simulator?
As the scenes here are entirely synthetic, they do not correspond to an actual location therefore it is not feasible to show a location map for reference (i.e., show the location being mapped on earth). See comment below regarding spatial resolution. We have added “Scene units are arbitrary” to Figs. 3, 4, 6.
Lake Hume and Heron Island: how the 100 m pixel size might affect the simulator performance? The spatial resolution seems too coarse for both applications; or: which is the pixel size of maps in Fig.4?
Based on your comment, we have reviewed all references to spatial resolution in the manuscript and discovered several inconsistencies that we have now corrected. Firstly, the units are arbitrary pixels based on the dimensions of the synthetic scene and not meters. This language error came in through early versions of a project report. Thus, features in synthetic scenes have been generated to create enough variability but do not represent the size of these features in nature. Secondly, the instrument sampling rate (rather than spatial resolution) was 20 units (or pixels), not 100 m as described in the paper. This means that the units (specified as m in the manuscript) do not have any correlation to the size of these features as we know in nature. Rather the scenes were constructed of arbitrary units based on variability in a scene with a given dimension.
In order to ensure this is clear to the reader, we have made the following corrections:
L132: “The spatial units in scene construction and sampling are arbitrary and therefore may not represent feature sizes in nature.”
L 229 – deleted “a ground sampling distance of 100 m”
Table A1: “1 m changed to 1 pixel”
Table 1: “Nominal ground sampling distance” changed to “Sampling resolution” and the value changed from “20 m” to “20”.
L281: reference to GSD has been changed to “sampling resolution”
Round 2
Reviewer 1 Report
I cannot accept the use 9f CH-l as symbol for chlorophyll-a in Figure 2. Please, change in Figure by chl-a.
Appendix B is mentioned several times before Appendix A. Please, use correct order.
CDOM appears as cdom and CDOM. Use either but always the same.
The discussion needs to be improved. Authors must discuss other similar applications and the advantages of their proposal.
Author Response
I cannot accept the use 9f CH-l as symbol for chlorophyll-a in Figure 2. Please, change in Figure by chl-a.
Figure has been updated accordingly.
Appendix B is mentioned several times before Appendix A. Please, use correct order.
Appendices have been renamed and ordered correctly.
CDOM appears as cdom and CDOM. Use either but always the same.
Corrected.
The discussion needs to be improved. Authors must discuss other similar applications and the advantages of their proposal.
We thank the reviewer for this comment. We have referenced 11 additional studies in the discussion and introduction to fully contextualize the study. We would like to stress that the Simulator is a prototype and therefore a detailed comparison of the features of this Simulator to other available alternatives was not attempted. We hope that this greatly improves the manuscript (note: numbers below do not correspond to reference numbers in the manuscript).
- Kerekes, J.P.; Landgrebe, D.A. Simulation of optical remote sensing systems.IEEE Trans. Geosci. Remote Sens., 1989, 27(6), 762-771.
- Han, S.; Kerekes, J. P. Overview of Passive Optical Multispectral and Hyperspectral Image Simulation Techniques. IEEE J. Sel. Top. Appl. Earth Obs. Remote Sens. 2017, 10(11), 4794-4804.
- Coppo, P.; Chiarantini, L.; Alparone, L. End-to-End Image Simulator for Optical Imaging Systems: Equations and Simulation Examples.Advances in Optical Technologies. 2013, 295950.
- Verhoef, W.; Bach, H. Simulation of hyperspectral and directional radiance images using coupled biophysical and atmospheric radiative transfer models.Remote Sens. Environ., 2003, 87(1), 23-41.
- Guanter, L.; Segl, K.; Kaufmann, H. Simulation of Optical Remote-Sensing Scenes With Application to the EnMAP Hyperspectral Mission. IEEE Trans. Geosci. Remote Sens., 2009, 47(7), 2340-2351.
- Verhoef, W.; Bach, H. Simulation of Sentinel-3 images by four-stream surface–atmosphere radiative transfer modeling in the optical and thermal domains.Remote Sens. Environ., 2012, 120, 197-207.
- Drusch, M.; Del Bello, U.; Carlier, S.; Colin, O.; Fernandez, V.; Gascon, F.; Hoersch, B.; Isola, C.; Laberinti, P.; Martimort, P.; Meygret, A. Sentinel-2: ESA's optical high-resolution mission for GMES operational services. Remote Sens. Environ., 2012, 120, 25-36.
- Nieke J.; Frerick J.; Stroede J.; Mavrocordatos, C.; Berruti, B. Status of the optical payload and processor development of ESA's Sentinel 3 mission. IEEE Int. Geosci. Remote Sens. Symp., 2008, IV, 427-430.
- Qi,; Xie, D.; Yin, T.; Yan, G.; Gastellu-Etchegorry, J. P.; Li, L.; Zhang, W.; Mu, X.; Norford, L.K. LESS: LargE-Scale remote sensing data and image simulation framework over heterogeneous 3D scenes.Remote Sens. Environ., 2019, 221, 695-706.
- Lyu, Z.; Goossens, T.; Wandell, B. A.; Farrell, J. Validation of Physics-Based Image Systems Simulation With 3-D Scenes. IEEE Sensors Journal, 2022, 22(20), 19400-19410.
- Auer, S.; Bamler, R.; Reinartz, P. RaySAR-3D SAR simulator: Now open source. IEEE Int. Geosci. Remote Sens. Symp., 2016, 6730-6733.
Other changes:
- Emphasise that the Simulator is a “prototype”
- Emphasise that it is related to phase 0/A studies
Reviewer 2 Report
I think this version could be accepted.
Author Response
We thank the reviewer. The following changes have been made:
Figure 2 has been updated.
Appendices have been renamed and ordered correctly.
We have referenced 11 additional studies in the discussion and introduction to fully contextualize the study. We would like to stress that the Simulator is a prototype and therefore a detailed comparison of the features of this Simulator to other available alternatives was not attempted. We hope that this greatly improves the manuscript (note: numbers below do not correspond to reference numbers in the manuscript).
- Kerekes, J.P.; Landgrebe, D.A. Simulation of optical remote sensing systems.IEEE Trans. Geosci. Remote Sens., 1989, 27(6), 762-771.
- Han, S.; Kerekes, J. P. Overview of Passive Optical Multispectral and Hyperspectral Image Simulation Techniques. IEEE J. Sel. Top. Appl. Earth Obs. Remote Sens. 2017, 10(11), 4794-4804.
- Coppo, P.; Chiarantini, L.; Alparone, L. End-to-End Image Simulator for Optical Imaging Systems: Equations and Simulation Examples.Advances in Optical Technologies. 2013, 295950.
- Verhoef, W.; Bach, H. Simulation of hyperspectral and directional radiance images using coupled biophysical and atmospheric radiative transfer models.Remote Sens. Environ., 2003, 87(1), 23-41.
- Guanter, L.; Segl, K.; Kaufmann, H. Simulation of Optical Remote-Sensing Scenes With Application to the EnMAP Hyperspectral Mission. IEEE Trans. Geosci. Remote Sens., 2009, 47(7), 2340-2351.
- Verhoef, W.; Bach, H. Simulation of Sentinel-3 images by four-stream surface–atmosphere radiative transfer modeling in the optical and thermal domains.Remote Sens. Environ., 2012, 120, 197-207.
- Drusch, M.; Del Bello, U.; Carlier, S.; Colin, O.; Fernandez, V.; Gascon, F.; Hoersch, B.; Isola, C.; Laberinti, P.; Martimort, P.; Meygret, A. Sentinel-2: ESA's optical high-resolution mission for GMES operational services. Remote Sens. Environ., 2012, 120, 25-36.
- Nieke J.; Frerick J.; Stroede J.; Mavrocordatos, C.; Berruti, B. Status of the optical payload and processor development of ESA's Sentinel 3 mission. IEEE Int. Geosci. Remote Sens. Symp., 2008, IV, 427-430.
- Qi,; Xie, D.; Yin, T.; Yan, G.; Gastellu-Etchegorry, J. P.; Li, L.; Zhang, W.; Mu, X.; Norford, L.K. LESS: LargE-Scale remote sensing data and image simulation framework over heterogeneous 3D scenes.Remote Sens. Environ., 2019, 221, 695-706.
- Lyu, Z.; Goossens, T.; Wandell, B. A.; Farrell, J. Validation of Physics-Based Image Systems Simulation With 3-D Scenes. IEEE Sensors Journal, 2022, 22(20), 19400-19410.
- Auer, S.; Bamler, R.; Reinartz, P. RaySAR-3D SAR simulator: Now open source. IEEE Int. Geosci. Remote Sens. Symp., 2016, 6730-6733.
Other changes:
- Emphasise that the Simulator is a “prototype”
- Emphasise that it is related to phase 0/A studies
Reviewer 3 Report
Thank you for your responses and clarification. I'm glad that one of my comment helped you to correct some inconsistencies. Thanks for your study.
Author Response

(The authors gave the same response as above.)

Round 3
Reviewer 1 Report
Thanks to the authors for addressing my comments